# Growing up with a Chronically Ill Family Member—The Impact on and Support Needs of Young Adult Carers: A Scoping Review

**DOI:** 10.3390/ijerph19020855

**Published:** 2022-01-13

**Authors:** Hinke M. van der Werf, Marie Louise A. Luttik, Alice de Boer, Petrie F. Roodbol, Wolter Paans

**Affiliations:** 1Research Group Nursing Diagnostics, Hanze University of Applied Sciences, 9714 CA Groningen, The Netherlands; m.l.a.luttik@pl.hanze.nl (M.L.A.L.); w.paans@pl.hanze.nl (W.P.); 2Faculty of Social Sciences, Vrije Universiteit Amsterdam, 1081 HV Amsterdam, The Netherlands; a.de.boer@scp.nl; 3SCP, The Netherlands Institute of Social Sciences, 2500 BD The Hague, The Netherlands; 4Faculty of Medical Sciences, University Medical Center Groningen, 9713 AV Groningen, The Netherlands; petrieroodbol@gmail.com; 5Department of Critical Care, University Medical Centre Groningen, 9713 GZ Groningen, The Netherlands

**Keywords:** informal care, young adult carers, growing up with care, impact, needs, support, scoping review

## Abstract

This scoping review provides an overview of the impact of growing up with a chronically ill family member on young adults (18–25 years old), and their specific needs. Young adults represent an important life stage involving a transition to adulthood, during which individuals’ family situations can affect their future. We searched relevant studies following the guideline of Arskey and O’Mailley’s methodological framework and the PRISMA statement guidelines for scoping reviews in PubMed, PsychInfo and reference lists to identify articles for inclusion. Studies from 2005 to 2020 were included in this review. Of the 12 studies, six qualitative studies, five quantitative studies and one mixed method study were included. Eight studies discussed the impact, including consequences at a physical and mental level, at their personal development and future perspectives, but also positive effects, such as being capable of organizing their lives. Four studies discussed the needs of young adult carers, including emotional needs, support needs with regard to stimulating autonomy (arising from internal conflicts) and developing their own identity, and the concerned attitude of involved professionals. An unambiguous definition of the target group and further well-designed research are needed to improve clarity about the role of support, so that future professionals can adequately address the needs and wishes of young adults who grow up with an ill family member.

## 1. Introduction

Living with or caring for a chronically ill family member or older person is known to affect the health and well-being of family caregivers [1]. While a wealth of literature describes the impact of caregiving on the health and well-being of spouses, partners and parents, less is known about the impact of care situations or caregiving on children and young adults [2]. Studies among young carers up to 18 years of age suggest that growing up with a chronically ill family member can result in stress, problems in the parent–child relationship, (social) development problems and diminished school results [3,4]. Additionally, school-aged children with a chronically ill family member appear to have a greater need for and use of help and support, compared with peers without a care situation at home [5]. Special attention needs to be paid to those youngsters who are in need of support but are not able to seek and find accessible help. An examination of policy responses for young carers aged 18 or younger underlined the importance of recognizing and raising awareness among both professionals and governments [6].

However, for the specific age group of 18 to 25 years old who grow up with (including those who care for) a chronically ill family member, there is no clear overview of the types of problems they encounter and what kind of support they need.

This is all the more striking because this is a period of ‘emerging adulthood’ [7]. This life stage is characterised by the development of autonomy and identity and the feeling of being between childhood and adulthood [8,9,10]. Many choices and life paths are open to individuals in this age group because they are not limited by many restrictions (parental supervision) and responsibilities (family and financial obligations) [11]. The natural process of developing one’s own identity with stable role models and breaking away from parental supervision seems to be complicated when emerging adults grow up with a chronically ill family member [12]. Growing up under these circumstances can be a risk factor for the health and (social) development of younger adults [3].

An overview of the scientific literature on the impact of young adults who are confronted with care or caregiving and their support needs could be helpful for a wide range of professionals employed by (local) governments who are tasked with contributing to the successful growth of young adults [6,13]. The aim of this study is therefore to provide an overview of the scientific literature describing (1) the impact of growing up with (including caring for) a chronically ill family member on young adults and (2) their specific support needs.

## 2. Method

As little is known about the impact of people aged 18 to 25 growing up with or caring for an ill family member and related support needs, we explored this issue in a scoping review. We chose this type of review because an initial search in the PubMed database revealed that the nature of the studies, their methodical approaches and the scientific content differed widely. To gain insight into the range of studies, we mapped and summarised the relevant literature, regardless of study designs, and identified gaps in existing literature [14,15,16].

Collection and reporting of data were performed following the guidelines of Arskey and O’Mailley’s [17] methodological framework and the PRISMA statement guidelines for scoping reviews [18], including a systematic search in the electronic databases PubMed and PsychInfo.

### 2.1. Database Search

The research was conducted between 16 February 2021 and 3 April 2021. We formulated a broad search strategy including several synonyms for the key concept of ‘Growing up with a chronically ill family member by young adults’ (see Figure 1). We tried to be as inclusive as possible by limiting the use of filters and limits in the search strategy in advance.

### 2.2. Inclusion Criteria

Inclusion and exclusion criteria were drawn up based on the focus of the research question. Studies were included if they (1) were written in English, (2) were published in a peer-reviewed journal and (3) contained a scientific study that qualitatively or quantitatively described the impact of caregiving and/or that described the needs of young informal carers. In the literature, the definition of young adult carers is based on age [19]. Following the criterion of 18–25 years old, according to Arnett’s [7] theory of emerging adulthood, studies among caregivers under the age of 16 years or with a study population with a mean age either <18 years or older than 25 years were excluded from our search, unless a clear distinction was made between the different age groups in the results paragraph. This means that existing well-designed studies [20,21] which included broader age groups with carers who were younger than 18 years were not included in this scoping review.

Moreover, apart from age, a broad definition of ‘care’ is used in the literature. On the one hand, many studies describe young carers up to 25 years old as adults who carry out tasks, often on a regular basis, assuming a level of responsibility for the well-being of their family members [22] On the other hand, there are also studies who focus on growing up with worries for a loved one [5,19]. Indeed, while young adults may live away from their family and therefore no longer perform specific tasks, they may (still) feel the impact of the care situation at home. In this study, we therefore use the definition of growing up with a chronically ill family member with a broad definition of care, regardless of whether or not young adults carry out tasks.

### 2.3. Study Selection

After the first screening of titles and abstracts, supported by the software program Rayyan [23], the full text of the selected articles was examined. While examining the included articles, two independent reviewers (HMW and MLL) systematically abstracted the focus, design, methodology, sample size, definition of the population, recruitment, level of evidence (Table 1) and general key findings concerning the impact on and support needs of young adults who are confronted with care or in a care situation (aged 18–25 years, Table 2).

### 2.4. Quality Assessment

We used the relevant Mixed Methods Appraisal Tool [24] and the Joanna Briggs Institute’s [25] critical appraisal checklists, fitting the different study designs. To ensure reliability, articles published by the first author (HMW) were assessed by an independent reviewer. Furthermore, levels of evidence were estimated. Papers that seemed to meet the inclusion criteria but caused doubt due to ambiguities were analysed once more by a third investigator (WP) until consensus was reached.

**Table 1 ijerph-19-00855-t001:** Matrix scoping review.

(Study Number) Author Details, Year	Title	Study Design, Study Location and Date Collecting Data	Age Range Participants	Number of Participants	Aim	Definition of the Population and Recruitment	Level of Evidence [25]
(1) Levine et al. (2005)	Young adult caregivers: A first look at an unstudied population.	Desk research based on existing data of 2 national surveys (from 1998 and 2004) of adult caregivers. USA.	18–25 years old	*n* = 234	Describing the population of young adult caregivers and laying the groundwork for future studies.	‘Young adults aged 18 to 25 years who are caregivers for ill, elderly, or disabled family members or friends’ (not explicitly described). Desk research based on existing data among adult caregivers in the USA.	3c
(2) Ali et al. (2012)	Daily life for young adults who care for a person with mental illness: a qualitative study.	Interviews and focus groups in 2008. Sweden.	16–25 years old	*n* = 23	Elucidate the daily life of young people who care for friends or family members with mental illness and explore how they manage in everyday life.	‘16 to 25 years old, supporting a close friend or family member who suffered from mental illness’. Recruitment via advertisements in newspapers, leaflets and a webpage.	4
(3) Ali et al. (2013)	Support for young informal carers of persons with mental illness: a mixed-method study.	Mixed method (interviews and self-administered questionnaire) in 2008–2009. Sweden.	16–25 years old	*n* = 235	Exploring how young informal carers of a person with a mental illness experience and use support.	‘16 to 25 years old, supporting a close friend or family member who suffered from mental illness’. Recruitment by searching in the Swedish national population register. A recruitment company screened prospective participants for eligibility.	4
(4) Greene et al. (2017)	The relationship between family caregiving and the mental health of emerging young adult caregivers.	Cross sectional survey in 2009. USA.	18–24 years old	*n* = 353 (81 past caregivers, 76 current/past caregivers and 196 non-caregivers).	Examination of the relationship of family caregiving responsibilities and the mental health and well-being of young adult carers.	‘18–24 years old young adult carers’ (not explicitly described). Recruitment by a survey among students.	3c
(5) Moberg et al. (2017)	Striving for balance between caring and restraint: young adults’ experiences with parental multiple sclerosis.	Interviews in 2014. Denmark.	18–25 years old	*n* = 14	Exploring and describing how young adults experienced growing up with a parent with multiple sclerosis and how these experiences continue to influence their daily lives.	‘Young adults between 18–25 years of age growing up with a parent with multiple sclerosis’. Recruitment via advertising in the MS magazine and website, Facebook groups and MS hospital clinics across Denmark.	4
(6) Boumans and Dorant (2018)	A cross-sectional study on experiences of young adult carers compared to young adult noncarers: parentification, coping and resilience.	Cross sectional survey in 2014/2015. Netherlands.	18–24 years old	*n* = 297 (56 carers and 241 non carers).	Exploring young adult carers’ perceptions of parentification, resilience and coping compared to young adult non carers.	‘Young adult carers aged 18–24 years’ (not explicitly described). Recruitment by surveys; students were approached through their mentors to complete a questionnaire during mentor class.	3c
(7) van der Werf et al. (2019)	Students growing up with a chronically ill family member; a survey on experienced consequences, background characteristics, and risk factors.	Cross sectional survey in 2017. Netherlands.	16–25 years old	*n* = 237	Exploring the consequences for young adult carers following bachelor or vocational education programs, and the influence of various background characteristics and risk factors.	‘Students (16–25 y) who identified themselves as growing up with a chronically ill family member’. Recruitment by a survey among students.	4
(8) Day (2019)	An empirical case study of young adult carers’ engagement and success in higher education.	Interviews (date of collecting not described). Australia.	18–25 years old	*n* = 12	Examination of the educational experiences among young adult caregivers.	‘18–25 y old young adult carers’. Recruitment via university (not explicitly described).	4
(9) Kettell (2020)	Young adult carers in higher education: the motivations, barriers and challenges involved—a UK study.	Interviews (date of collecting not described). UK.	20–23 years old	*n* = 3	Understanding the lived experiences of young adult carers who are in higher education.	Definition population not explicitly described. Recruitment via a flyers displayed at various locations around the university campus.	4
(10) van der Werf et al. (2020)	Experiences of Dutch students growing up with a family member with a chronic illness: A qualitative study.	Focus groups in 2017/2018. Netherlands.	18–25 years old	*n* = 25	Describing the themes experienced by students growing up with a chronically ill family member.	‘Young adults growing up with a chronically ill family member’ (not explicitly described). Recruitment by a survey among students. Students with a chronically ill family member were asked if they were willing to participate in a focus group.	4
(11) van der Werf et al. (2020)	Expectations and prospects of young adult caregivers regarding the support of professionals: a qualitative focus group study.	Focus groups in 2017–2018. Netherlands.	18–25 years old	*n* = 25	Investigate the expectations and prospects of young adult caregivers regarding support from professionals to manage their own health and wellbeing.	‘Young adults growing up with a chronically ill family member’ (not explicitly described). Recruitment by survey among students. Students growing up with a chronically ill family member were asked if they were willing to participate in a focus group.	4
(12) Haugland et al. (2020)	The Burden of Care: A National Survey on the Prevalence, Demographic Characteristics and Health Problems Among Young Adult Carers Attending Higher Education in Norway.	Cross sectional survey in 2018. Norway.	18–25 years old	*n* = 41,205 (2220 carers compared to 38,985 non carers).	Examination the prevalence, characteristics and health outcomes among young adults who provide informal care to family members or others with physical or mental illnesses, substance misuse or disabilities.	‘Young adults (18 to 25 years) who provide informal care to family members or others with physical or mental illnesses, substance misuse or disabilities’. Recruitment by The SHoT2018 study (Students’ Health and Wellbeing Study) a national student survey for higher education in Norway.	3c

**Table 2 ijerph-19-00855-t002:** Impact and needs of young adult caregivers described in the articles.

Impact				Needs		
(Study Number) Author Details, Year	Physical Impact	Emotional Impact	Impact on Development and Future	Type of Support	Emotional Needs	Attitude Professional
1. Levine et al. (2005)	X		X			
2. Ali et al. (2012)		X	X			X
3. Ali et al. (2013)		X		X	X	X
4. Greene et al. (2017)		X				
5. Moberg et al. (2017)		X	X			
6. Boumans and Dorant (2018)		X				
7. van der Werf et al. (2019)		X				
8. Day (2019)	X	X	X			
9. Kettell (2020)		X	X	X		X
10. van der Werf et al. (2020)		X	X			
11. van der Werf et al. (2020)		X			X	X
12. Haugland et al. (2020)	X	X				

## 3. Results

Our search returned, after removing duplicates, 3035 titles. Based on titles, we excluded 2840 studies that discussed younger carers <18 years of age, older caregivers >25 years of age, parents caring for their children or non-English-language studies. In addition, based on abstracts, 67 studies with no clear distinction between age groups, without a substantial research-based methodology, discussing younger carers <18 years of age, older caregivers >25 years of age or focusing on professionals were excluded. In total, 128 articles were retained for full analysis, and two studies, identified through snowball sampling, were added after full analysis. In total, 12 articles were included based on the inclusion and exclusion criteria (see Figure 2).

## 4. Study Characteristics

This scoping review yielded 12 articles from seven countries. Two of these studies were conducted in the United States; two in Sweden; one each in Norway, Denmark, Australia and the UK; and four in the Netherlands.

Half of the 12 included studies (*n* = 6) adopted a qualitative design (see Table 1), and the findings were based either on in-depth or (semi-) structured interviews (*n* = 4) or on focus groups (*n* = 2). The number of respondents in the qualitative studies varied between 3 and 25 participants. Furthermore, one study with a mixed methods design and five quantitative studies were included. Remarkable within the studies are the different definitions of the target group, the various types of chronically ill persons the participants cared for, the scant information about the living situation of their participants and a gender imbalance. Except for the age bracket, seven studies did not explicitly describe the definition of their target group. In some studies (*n* = 8), the care recipient was a parent or a sibling of the caregiver; other studies also included young adult caregivers caring for a close friend or others (*n* = 4). In almost all included studies (*n* = 9), respondents provided informal care to family members with different problems or diseases. Other studies were more specific and only included young adult caregivers caring for a person with a mental illness (*n* = 2) or multiple sclerosis (*n* = 1). Two studies provided residential information regarding the living situation of their participants. In the remaining studies, the authors did not explicitly describe whether participants were living with their family members or on a campus, nor did they explicitly describe residential information about the family members’ living situation. Lastly, most studies noticed a gender imbalance, with a majority of female caregivers in their study population.

### Quality Assessment

After estimating the level of evidence of the design (Table 1, last column), the quality of the selected studies was measured with fitting checklists. Four qualitative studies [26,27,28,29] completely fulfilled the checklist criteria, scoring 10 out of 10 on the JBI Critical Appraisal Checklist for Qualitative Research [25]. In the remaining two qualitative articles, we found missing information or unclear information about the ethics and influence of the researcher on the study. Moreover, of the five analysed quantitative studies, four [12,30,31,32] almost completely fulfilled the checklist criteria, scoring 7 out of 8 on the JBI Critical Appraisal Checklist for Cross-Sectional Studies. Missing information was found in one or more of the investigated aspects among detailed study subjects, setting and/or using valid instruments. The mixed methods study completely fulfilled the checklist criteria of the MMAT [24].

## 5. Impact

Based on the papers included, three subthemes were defined by the authors to synthesise the impact of caregiving or a care situation on young adults: physical effects, emotional effects and effects on the development and future of young adults (see Table 2).

First, three studies describe the physical impact of growing up with a chronically ill family member. This impact is described as intensive [30]. Moreover, there is variance in the number of hours of care provided [30], and the number of hours spent on caring are associated with the magnitude of somatic symptoms, such as tiredness or insomnia [32,33].

Second, the included studies describe the emotional impact of caregiving on young adult caregivers in terms of mental health problems [12,26,27,28,29,32,34,35,36]. According to six studies, young adult caregivers experience stress by feeling worried about their family [26,28,29,34,35] and the impacts of the illness on their family member [26,27,28,34]. In three studies, it was reported that young adults feel alone [28,34,35]. because they are unable to express their feelings and desires to protect their family or wish to avoid burdening them with their sadness, questions, or problems. Moreover, in four studies, it was reported that reactions from others could hurt the feelings of young adults [26,28,34,35], thus making them feel alone or alienated from their peers [26,34]. In six studies, it was addressed that young adult caregivers felt responsible [26,28,29,31,33,34] and mentioned that their relationships with their family member were meaningful to them [27,28,29,30,31,32,33,34,35]. In four studies, it was found that these close relationships could result in conflicts [26,28,29] or guilt [28,29,33] among those caregivers at the thought that they did too little to help their parents.

Three studies describe the physical and emotional impact of caring, comparing young adult carers with non-carers [12,31,32]. Insomnia and symptoms of depression and anxiety were found to be significantly more prevalent among young adult carers compared to non-carers [12,32], although a positive effect of caregiving, such as a higher score for emotion-focused coping [31], was also found among young adult carers compared to non-carers. Greene and colleagues [12] suggested that past caregivers could use their coping skills more efficiently compared to caregivers because they no longer had to deal with the daily stress of caring.

Third, there were studies that underlined the impact on personal development and future perspectives. In one paper, positive effects were described: young adult caregivers were more advantaged in terms of organizing their educational and private lives [28]. It is unclear how these positive effects relate to their personal development and future perspectives.

Three studies identified that young adult caregivers often have less time for social activities, such as hobbies, and time to see their friends [29,30], which could result in peer alienation in an age bracket where social contacts are important [26]. In other studies, some young adult carers avoided school when they felt misunderstood by others; this could also have consequences for their future perspectives [33,34]. Others were missing classes because they had to take care of their family [29,33]. Furthermore, young adult caregivers experience internal conflicts regarding whether to start their own life and career or whether to remain close to their family and childhood homes in order to help [26,28,33]. As a result, young adult caregivers may not opt for studies, an internship or a career abroad [26,33], and they are more likely to enrol in health-related education [28].

## 6. Needs

Four publications were found studying the needs of young adult caregivers [27,29,34,35]. Three themes were addressed in these publications on the emotional needs of young adults, the attitudes of professionals and the types of support. In some studies, young adult caregivers described the need to be involved in caring for their chronically ill family member [27,34] and indicated that they need someone to talk to in times of crisis [35]. Flexibility (e.g., at their school or university), whereby they can easily ask for special leave to help their family, was found to be useful [29]. These individuals ask for a professional who takes their needs and the needs of their family members seriously [27,34,35]. Moreover, young adult caregivers ask for professionals who consider them as having an important role in the care situation [27,34], they should listen to them [27,29,34,35], encourage to share their situation [35] and help with specific coping strategies for handling their family situation [27,35]. Young adult caregivers express they need for information about their family member’s illness, the consequences thereof, and the symptoms [27]. They prefer to search online for this information as well as for the particulars about care providers who can offer support [35]. Some studies also mentioned face-to-face contact in combination with web support [29,35] as being useful. In addition to professional help, these authors underlined the importance of peer groups for sharing daily life experiences and coping strategies which can support these young adult caregivers. None of the included studies in this scoping review compare the support needs of young adult caregivers with non-carers.

## 7. Discussion

With this scoping review, we aimed to study what is known about young adults, the impact of growing up with a chronically ill family member, and the support they need. It was observed that the definition of ‘young adult caregivers’ or ‘growing up with illness’ differs across studies, making it difficult to define this target group (apart from age). We therefore performed a broad search and non-stringent inclusion and exclusion criteria resulting in 12 appropriate studies. This review shows that informal care seems to have an impact on the physical and mental condition of young adult informal carers and on their personal development and future prospects. There are different types of support to help young adult carers with their emotional needs. The attitude of a professional is also important in identifying and interacting with these young adults and meeting their needs.

Young adult carers (aged 18–25) appear to mention a physical impact more often than younger carers (<18 years old), who are regularly seen with internalizing problem behaviour such as psychosomatic complaints [37]. This finding has also been established in various studies among caregivers older than 25 [38,39]. Young adults compare themselves to their peers’ environments. Therefore, they might be more aware of the effect of their caring on their health and well-being and feel the physical toll of caring [7,40].

The included studies indicate a variety of mental problems arising from growing up with a chronically ill family member. These problems occur in younger caregivers (<18 years old) [4,41]. Other studies focussing on young caregivers describe mental problems as well as externalizing (aggressive and delinquent behaviour) and internalizing (depressive symptoms, anxiety, withdrawal) behaviour [3,37,42]. Studies on older caregivers (>25 years old) [43] relate to stress, depressive and anxiety symptoms. Because no comparison has been made with other age groups, it is not clear whether or not the 18–25-year-olds who grow up with care deviate with regard to the occurrence of mental problems. Furthermore, young adult carers described in this scoping review mentioned some positive effects of caregiving, such as organizing their educational and private lives. These effects have also been found among younger caregivers, who seem to be often capable of carrying and organizing an increasing amount of responsibility as they age [44]. Dearden and Becker [45] and Heyman and Heyman [46] ascertained that young caregivers (<18 years old) generally become more mature and learn to take on more responsibility compared to peers who have not grown up with a family member with a chronic illness.

The impact of caring for such a family member on young adult carers’ personal development and future perspectives is characterised as having less time for social activities and seeing friends, avoiding school, and experiencing internal conflicts regarding choices about their future and helping their family. Additionally, younger caregivers (<18 years old) experience social restrictions and have high levels of absenteeism at school [21,47]. Internal conflicts appear to be specifically recognised by young adult carers, which can be explained by their developmental stage. The caregiving situation in their family can take a central role in their lives, which contrasts with the normal process of emerging adulthood, whereby young adults focus on becoming autonomous and developing their own identity [8,9,10].

Regarding young adult carers’ support needs, we found two studies describing the importance and types of support (face to face- support versus online support and peer groups). Both help from professionals and/or informal help from peers are needed by young adult carers. The types of support required seem partly similar to those needed by younger caregivers [48,49,50]: being a young carer is experienced as a process of identity formation, which explains their wish to be involved in the care of their chronically ill family member. It is also partly different because of the transition from childhood to adulthood can increasingly conflict with their care identity. To separate from this role and develop their own personal identity, young adult carers hence need specific coping strategies to become autonomous and to deal with conflicts, anxiety and stress that arise.

According to young adult carers, professionals must consider that they are often well aware of their situation and in need of someone who is flexible, easy to reach and able to aid them in the process of developing their identity and independence in combination with their caregiving role. The transition to their own personal identity requires a different attitude from a professional towards young adult carers. All young carers, despite their age, are often overlooked by professionals and must thus frequently ask for help themselves [6,13]. This can be challenging, especially at an age when they are expected to become independent and autonomous and where they do not want to be different from their peers by seeking professional support [7].

## 8. Future Research

This review contains mainly qualitative research with small and selected samples or cross-sectional studies without a control group, making generalization difficult. Existing well-designed quantitative studies [20,21] with larger datasets often included broader age groups with young carers (<18 years) without published data about various age groups; for this reason, they are not included in this paper. Further well-designed research is thus still highly needed with enough respondents aged 18 to 25 years, representative samples and control groups to determine the extent and nature of the impact on (future adult) life and possible support interventions. Intervention studies in which young adult caregivers contribute ideas, develop and evaluate, such as in international research and innovation we present for younger informal caregivers [51], could be a possible next step in research into this target group. Lastly, 9 out of the 12 included publications in this scoping review contain more female caregivers than male. This gender imbalance is also found in other studies among older caregivers [52,53]. It is not clear if female caregivers are more inclined to identify themselves as being a (young adult) caregiver or whether they feel more able or comfortable, for example, to discuss the health of their (chronically ill) family members. The role of professionals in supporting young adults apart from peer-support is not discussed in this study. Future research into possible professional assistance from childhood to adulthood seems to be essential to develop adequate and effective support, to prevent serious long-term problems for young (adult) caregivers and to enable them to develop as optimally as possible.

### Strengths and Limitations

A strong point of this review is the broad scope of focus on a target group that has not yet been extensively researched. In this review, we attempted to paint a picture of the specific impact on young adults and their needs while growing up with a chronically ill family member. A search strategy was applied that included studies regardless of the chosen methodology and level of evidence. We searched for articles on informal caregivers and care situations in the age group of 18–25 years old, the period of emerging adulthood [7]. Studies in which the distinction between young carers (young children) and young adult carers was unclear or insufficiently explicit to properly interpret the results in the light of our research question were excluded from this review. Some articles with participants between the ages of 16 and 18 were also included in our review [34,35,36], but only when the mean age of the total study population was >18 years. The variation in the methods used means that generalization of the results is often not possible without further processing. A limitation of this study is that the researchers could only include English-language studies. The research group did not have budget to enable extensive translations into other languages.

## 9. Implications

Our findings suggest that young adult caregivers require specific attention in the development of care policies. The use of an unambiguous definition in line with the literature on emerging adulthood could be a step towards more specific awareness and research regarding this target group [33].

Aside from a clear definition, interdisciplinary and inter-professional collaboration in supporting these young adults is key [2]. Due to their age, young adult caregivers find themselves in different roles within society, making it difficult to designate a professional who can provide support and take responsibility for coordinating interdisciplinary and inter-professional collaboration among professionals.

Due to the lack of clarity when asking for support from a professional or peers, what support these young adults could mean is not clear yet. When young adult carers are still participating in education, the educational institution may play a major role in supporting these students [29,33]. Previous research on young adult carers shows that contacting peers in a supportive network [54], taking measures to support them by acknowledging their situation, and offering flexible solutions to practical problems may help to improve their quality of life [27,29,33]. Intervention studies are needed to design targeted interventions including young adult with and without caregiving tasks. Furthermore, health care professionals could play an important role by paying attention to their patients’ family situation. Professionals should view family members as not only crucial in supporting the ill family member, but also as potentially vulnerable in the care situation [27,34]. We have chosen to perform a scoping review, with the aim of providing a broad inventory of relevant literature. This review shows that studies centring on concrete interventions that actually support this target group of young adults are still very limited. It is recommended to look in particular at the effectiveness of concrete interventions, so that in the long term, an overview of group comparisons and before and after studies can be provided.

## 10. Conclusions

Young adults aged 18–25 who grow up with care or provide care suffer not only on a physical and mental level, but also in terms of their personal development and future perspectives. They have specific support needs to enable them to become autonomous and develop their own identity compared to younger carers. More well-designed research with representative samples and control groups is necessary to determine the extent and nature of the impact of this care-related role on the lives of young adult carers and possible interventions. Furthermore, research is required to build knowledge on the support of young adults in such a way that they make a successful transition into adulthood.

## Figures and Tables

**Figure 1 ijerph-19-00855-f001:**
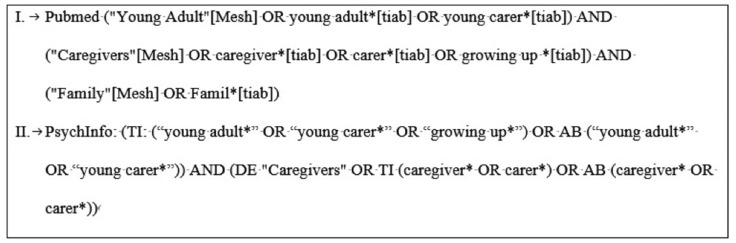
Database search.

**Figure 2 ijerph-19-00855-f002:**
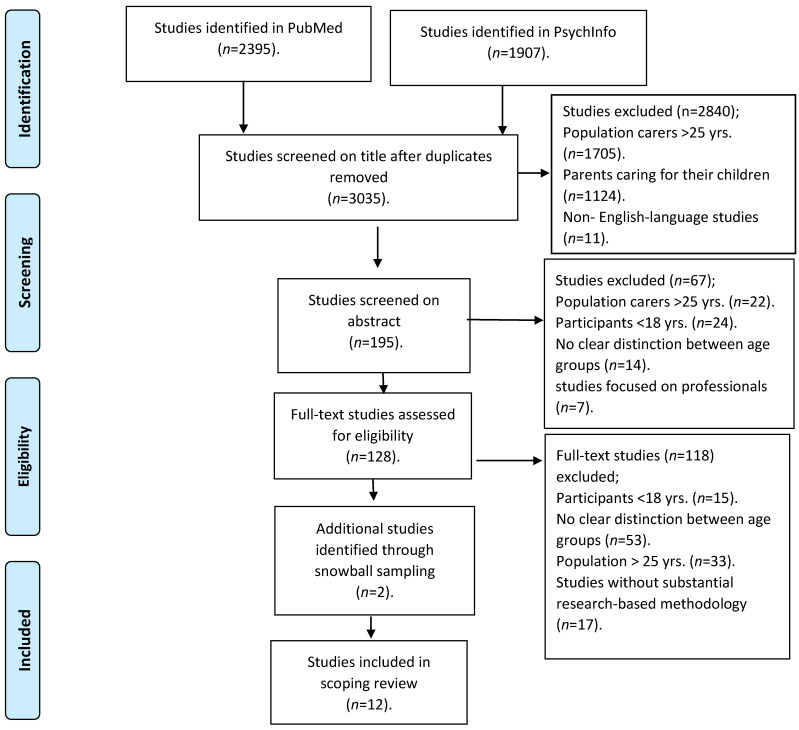
Flow chart; search strategy and number of records identified.

## Data Availability

No new data were created or analyzed in this study. Data sharing is not applicable to this article.

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
