# Peer review of "Growing up with a Chronically Ill Family Member—The Impact on and Support Needs of Young Adult Carers: A Scoping Review"

_ijerph, 2022, doi:10.3390/ijerph19020855_

Round 1

Reviewer 1 Report

I am sorry to see that they have only paid attention to articles published in English, which although they are the vast majority, there are other languages such as Spanish and French that can provide great studies, so I think it generates an important bias in their study.
On the other hand, the flow chart is blurred, probably due to the conversion of the image, but this would have to be checked.

Author Response

Dear reviewer, thank you for your time and effort to review our paper.

1.We agree that there are other important non-English studies that can provide new insights into this topic. Although, with this review we want to encourage researchers to also publish their work in English in order to reach a wider audience and thus share all available knowledge among this target group but this is also a limitation.

We added the following sentence in the limitations paragraph: ‘ A limitation of this study is that the researchers could only include English-language studies.  The research group did not have budgets to enable extensive translations into other languages.’

2.Thank you for pointing out the blurred flow chart. We will check it again to make sure it is correctly displayed.

Reviewer 2 Report

Dear Authors,

congratulation on excellent work. I have just a few corrections, as follow:

In summary you state that of the 12 studies included, 6 used a qualitative research design. What is the design of the remaining 6? Points are missing in several places in the text of the manuscript after the reference, please add points. In the introduction, quote the sentence: "Concerning people aged 18 and 25 who grow up with a chronically ill family member, it is unclear what types of problems they encounter and what kind of support they need." However, in the rest of the Introduction you describe the problems that occur in young people who have grown up with a chronically ill family member. Please format this sentence according to the meaning of the rest of the text. In the beginning Method the word is repeated twice. You have presented the results of the literature review in detail, clearly and concretely, given the diversity of sample definition. In conclusion, I think that the work is really interesting and that the work is given the difficulty of defining the sample, as well as defining all the difficulties that young people who care for a chronically ill family member have. Also, I consider it extremely important that you pointed out the need for as soon as possible and as concrete studies as possible in order to accurately determine the above and to find adequate solutions to help young people. 

Author Response

Dear reviewer, thank you for your kind words and helpful suggestions to improve our paper.

1.Thank you for pointing out the unclarity in the abstract and the missing points. We followed your suggestion and have rephrased the result section in the abstract with: ‘Of the 12 studies, six qualitative studies, five quantitative studies and one mixed method study were included.’

2. Furthermore, we checked our manuscript, added the missing points and removed the double word in the method.

3. We changed the sentence: ‘Concerning people aged 18 and 25 who grow up with a chronically ill family member, it is unclear what types of problems they encounter and what kind of support they need.’ into: ‘However, for the specific age group of 18 to 25 years old who grow up with (including those who care for) a chronically ill family member, there is no clear overview of  the types of problems they encounter and what kind of support they need.’

Reviewer 3 Report

The topic of the manuscript “Growing up with a chronically ill family member – the impact on and support needs of young adult careers: A scoping review” is of interest to IJERPH readers. Some comments are provided in order to help to improve the work:

Introduction:

In general, the introduction provides a concrete context for the topic to be addressed. However, it is necessary for the authors to check spelling and punctuation, for example in line 2 of the introduction (just before reference 1).

Methods:

The authors shall specify the date on which each source of information was searched or last consulted, as well as the filters and limits set for the search strategies provided.

Results:

The authors are highly recommended to considerably improve the quality of the figures and tables provided, especially the flow-chart. It is recommended to review the recommendations of the PRISMA guide and provide information from the beginning of the search (4302 titles) and indicate the reason for rejection of each one of them, as well as the number of titles excluded for each reason. For example, how many titles were excluded for including participants < 18 years old? All this information is not clear and transparent, preventing the reproducibility of the review.

Regarding the tables, it is suggested that the authors organize the information/headings according to some kind of characteristic, e.g. by year of publication, and that they provide key, easy-to-read information in the tables, even if this means enlarging the table and including individual columns for specific characteristics such as: age range of participants, number of participants, etc.

Discussion:

It is advisable that the authors include the overall results of the review in the first paragraph and discuss them following the structure used in the objectives and results, i.e. start with the impact and continue with the needs. This shall be compared with the findings of similar studies.

It is also important that the authors discuss any limitations of the evidence included in the review and the limitations of the review processes used.

Financing:

The authors are recommended to describe the sources of financial or non-financial support for their review, and the role of funders or sponsors in the review.

Author Response

Dear reviewer, thank you for these helpful suggestions to improve our paper.

1/2. Thank you for pointing out the missing information about the search date. For more clarity, we added the following sentence: ‘The research was conducted between 16-2-2021 and 3-4-2021.’ Furthermore, we have tried to be as inclusive as possible by limiting the use of filters and limits in the search strategy in advance. We added this information in the main text. Lastly, we checked our manuscript for spelling and added the missing points.

3/4.We improved the quality of our tables and added specific information to the flowchart for more clarity. Furthermore we followed your suggestions to organize the tables by year of publication and added individual columns for a more easily readable segment.

5.We agreed with your suggestion to include the overall results of the review in the first paragraph of the discussion and added the next paragraph:

This review shows that informal care seems to have an impact on the physical and mental condition of young adult informal carers and on their personal development and future prospects. There are different types of support to help young adult carers with their emotional needs. The attitude of a professional is also important in identifying and interacting with these young adults and meeting their needs.

Furthermore, thank you for pointing out the missing information about limitations and funding.

6.We added the following paragraphs:

‘Strength and limitations

A strong point of this review is the broad scope on a target group that has not yet been extensively researched. In this review, we have attempted to paint a picture of the specific impact on young adults and their needs while growing up with a chronically ill family member. A search strategy was applied that included studies regardless of the chosen methodology and level of evidence. We searched for articles on informal caregivers  and - care situations in the age 18 - 25 old, the period of emerging adulthood [7]. Studies in which the distinction between young carers (young children) and young adult carers was unclear or insufficiently explicit to properly interpret the results in the light of our research question were excluded from this review. Some articles with participants between the ages of 16 and 18 were also included in our review [34-36], but only when the mean age of the total study population was >18 years. The variation in the methods used means that generalization of the results is often not possible without further ado. A limitation of this study is that the researchers could only include English-language studies. The research group did not have budgets to enable extensive translations into other languages.’

7.‘Funding

This study was funded by the university of applied sciences of Groningen and the university of Groningen as part of a PhD-study. No additional funds have been used.’

Reviewer 4 Report

Considerations:
Methodology
The research question is not adequately described and it is not specified which system has been 
used to construct the search equation (PICO or SPC system).
Likewise, I do not see in the inclusion criteria those related to temporality: the period of time 
covered by the systematic review. 
Very old articles are detected in the bibliography and it is not specified whether they have been 
included because of their importance.
Nor does it specify whether systematic reviews have been carried out on this topic, although the 
bibliography does include the following
Introduction
I consider that the introduction is excessively poor, statements are made without being 
accompanied by solid arguments such as "There is a lack of an overview of the scientific 
literature on the impact of young adults facing care or caregiving and their support needs" and 
it relies on outdated literature and on populations that are not the subject of the study

Author Response

Dear reviewer, thank you for your time and effort to review. Your suggestions are helpful to improve our paper. We have tried to follow the recommendations that were made.

1.This scoping review is an overview that is broader than looking at specific interventions and strategies that support this target group. Therefore, the research question is broadly described following the PRISMA-scoping review Checklist and the Arksey and O'Malley methodological framework for conducting a scoping study instead of a PICO or SPC system.

2/3. Studies from 2005 to 2020 are included in this review. The older studies (Arnett and Furstenberg) are included in the bibliography. These studies have been used in particular because they are founding fathers of the theory on emerging adulthood.

Thank you for pointing out the missing information about the search date. For more clarity, we added the following sentence: ‘The research was conducted between 16-2-2021 and 3-4-2021.’

4/5. There are indeed reviews for age groups <18 or >25, but to our best knowledge we could not find a review specifically addressing our target group of 18-25 year olds. We agreed that we could describe this more carefully and added the following sentences for more clarity: ‘However, for the specific age group 18 to 25 years olds who grow up with (including those who care for) a chronically ill family member, there is no clear overview of the types of problems they encounter and what kind of support they need.’

 ‘An overview of the scientific literature on the impact of young adults who are confronted with care or caregiving and their support needs could be helpful for a wide range of professionals of (local) governments who are tasked with contributing to the successful growth of young adults [6,13].’